# Ocular Surface Side Effects of Novel Anticancer Drugs

**DOI:** 10.3390/cancers16020344

**Published:** 2024-01-13

**Authors:** Livio Vitiello, Filippo Lixi, Giulia Coco, Giuseppe Giannaccare

**Affiliations:** 1Eye Unit, “Luigi Curto” Hospital, Azienda Sanitaria Locale Salerno, 84035 Polla, SA, Italy; livio.vitiello@gmail.com; 2Eye Clinic, Department of Surgical Sciences, University of Cagliari, 09124 Cagliari, CA, Italy; f.lixi1@studenti.unica.it; 3Department of Clinical Sciences and Translational Medicine, University of Rome Tor Vergata, 00133 Rome, RM, Italy; giulia.coco@uniroma2.it

**Keywords:** ocular surface, drugs, cancer, dry eye, side effects

## Abstract

**Simple Summary:**

Anticancer drugs can determine alterations in the ocular surface system as side effects that can be referred to by patients as ocular discomfort symptoms and, sometimes, as reduced visual acuity. For this reason, it becomes essential to spread knowledge of the entire spectrum of ocular side effects of these drugs, especially for newer and more recent drugs. This review aims at analyzing the adverse effects on the ocular surface of these innovative anticancer drugs, trying to highlight the physiopathogenetic mechanisms underlying these complications and providing useful indications to clinicians for their proper management.

**Abstract:**

Surgery, anticancer drugs (chemotherapy, hormonal medicines, and targeted treatments), and/or radiation are common treatment strategies for neoplastic diseases. Anticancer drugs eliminate malignant cells through the inhibition of specific pathways that contribute to the formation and development of cancer. Given the ability of such pharmacological medications to combat cancerous cells, their role in the management of neoplastic diseases has become essential. However, these drugs may also lead to undesirable systemic and ocular adverse effects due to cyto/neuro-toxicity and inflammatory reactions. Ocular surface side effects are recognized to significantly impact patient’s quality of life and quality of vision. Blepharoconjunctivitis is known to be a common side effect caused by oxaliplatin, cyclophosphamide, cytarabine, and docetaxel, while anastrozole, methotrexate, and 5-fluorouracil can all determine dry eye disease. However, the potential processes involved in the development of these alterations are yet not fully understood, especially for novel drugs currently available for cancer treatment. This review aims at analyzing the potential ocular surface and adnexal side effects of novel anticancer medications, trying to provide a better understanding of the underlying pharmacological processes and useful insights on the choice of proper management.

## 1. Introduction

Cancer is a group of neoplastic disorders characterized by the abnormal and uncontrolled division and proliferation of cells in the body. The incidence of cancer is expected to increase, with projections estimating it to reach 23.6 million cases worldwide by 2030, making it one of the main causes of mortality. Despite this figure, overall cancer survival has improved over the last few decades, thanks to the efforts in cancer screening systems, early diagnosis, and development of new treatments [1]. In this context, novel medications are constantly being developed in the hope of efficiently reversing the course of the disease or averting further negative consequences. Concerning anticancer therapies, new and innovative drugs have evolved to become more specifically focused on one or multiple aspects of the disease pathophysiology. Novel chemotherapeutic agents, like S-1, immune checkpoint inhibitors in cancer immunotherapy, fibroblastic or epidermal growth factor receptor inhibitors, and an ever-expanding array of monoclonal antibodies that target neoplastic conditions (e.g., bentamab mafodotin for multiple myeloma), are only some of the newest developed anticancer drugs. Even though medication prescription is a recognized risk factor for ocular surface disorders [2], the onset and diagnosis of ocular surface side effects such as dry eye disease (DED) and blepharoconjunctivitis typically take some time.

The last Dry Eye Workshop II (DEWS II) report on iatrogenic DED mainly focused on commonly used drugs, like corticosteroids, nonsteroidal anti-inflammatory drugs, antihistamines, antidepressants, and antihypertensives [3], providing limited information on anticancer drugs, especially the more modern ones. The aim of this review is to report an in-depth overview of possible ocular surface and adnexal side effects of newly introduced anticancer drugs, trying to better understand the pharmacological mechanisms underlying the occurrence of these complications and providing useful insights on their correct management.

## 2. Materials and Methods

We carried out a search on the PubMed medical database. The sentences “ocular surface” and “ocular adnexa” were used in conjunction with a number of other “text words” pertaining to anticancer medications (such as “immune checkpoint inhibitors”, “epidermal growth factor receptor inhibitors”, “fibroblast growth factor receptor inhibitors”, “human epidermal growth factor 2 inhibitors”, “mitogen-activated protein kinase inhibitors”, “proto-oncogene B-Raf inhibitors”, “selective estrogen receptor modulators”, “breakpoint cluster region-Abelson inhibitors”, “aromatase inhibitors”, “FMS-like tyrosine kinase 3 inhibitors”, “Bruton tyrosine kinase inhibitors”, “anaplastic lymphoma kinase inhibitors”, “vascular endothelial growth factor receptor inhibitors”, “proteasome inhibitors”, “antibody–drug conjugates”, “novel anticancer drugs”) AND possible side effects on the eyes (such as “blepharoconjunctivitis”, “keratitis”, “blurred vision”, “itchy eye”, “corneal edema”, “conjunctivitis”, “keratoconjunctivitis”, “conjunctival hyperemia”, “ocular pain”). Text terms were selected considering the current literature as well as information from relevant bibliographies. The search was concluded in October 2023, with the earliest publication date set in January 1990.

No language restrictions were applied to searches; nevertheless, only English-language articles and reviews were examined. Moreover, manual searches from the original results were also conducted to find further bibliographic inclusions.

## 3. Results

### 3.1. Immune Checkpoint Inhibitors

In recent years, immunotherapy has emerged as a cutting-edge treatment for several malignancies, and immune checkpoint inhibitors are currently used to treat lymphoma, melanoma, and lung and pancreatic cancers. Immune checkpoint inhibitors are able to block immunological checkpoint molecules, such as programmed cell death protein 1 (PD-1), cytotoxic T-lymphocyte-associated protein 4 (CTLA-4), and programmed death ligand 1 (PD-L1), and the goal of their inhibition is to boost the T lymphocytes’ immunological response against tumor cells. In 2011, the Food and Drug Administration (FDA) authorized the first immune checkpoint inhibitor, ipilimumab (anti-CTLA-4), for the treatment of metastatic or incurable cutaneous melanoma [4]. Avelumab (anti-PD-L1), durvalumab (anti-PD-L1), nivolumab (anti-PD-1), pembrolizumab (anti-PD-1), atezolizumab (anti-PD-L1), and, more recently, cemiplimab (anti-PD-1) have all been authorized since then [5,6,7,8,9,10]. As these drugs have shown a survival benefit in numerous cancers, including Hodgkin lymphoma, colorectal cancer, gastric cancer, urothelial carcinoma, metastatic non-small-cell lung cancer, hepatocellular carcinoma, and head and neck squamous cell carcinoma; the list of their FDA-approved indications and of their off-label use has expanded over time [11]. Regretfully, undesirable systemic immune system adverse events may arise as a result of their widespread and nonspecific immunological activity. Ocular immune-related adverse events are reported to occur in about 1–2.8% of patients receiving immune checkpoint inhibitors [12], resulting, therefore, less commonly than systemic immune-related adverse events. This is consistent with the role of the eye as an immune-privileged site. The most common immune checkpoint inhibitor-related ocular side effect is DED, with an estimated frequency ranging from 1.2% to 24.2%, based on the type of immune checkpoint inhibitor used [13]. The occurrence of DED does not often require the interruption of the treatment; however, it should be handled with specific ocular therapies that include tear substitutes, punctal occlusion, and topical cyclosporine, among others. Blepharitis and conjunctivitis are additional ocular surface side effects that can be conservatively managed without stopping the anticancer treatment [14,15,16,17,18,19,20,21,22]. Furthermore, several episodes of ulcerative keratitis, cicatrizing conjunctivitis, immune rejection of corneal transplant, and corneal perforation have been linked to the use of these drugs [23,24,25,26,27,28,29,30,31,32,33,34,35,36]. These complications are usually managed with topical or systemic corticosteroid medications; corneal perforations may be managed with conservative treatments, like bandage contact lenses, corneal gluing, topical autologous serum [37,38], tear substitutes, and punctal occlusion, for optimizing the ocular surface, or surgery (e.g., penetrating keratoplasty) in more severe cases [39]. Depending on the disease severity and individual response to ocular treatment, some circumstances may require stopping the anticancer therapy [13].

### 3.2. Epidermal Growth Factor Receptor Inhibitors

The epidermal growth factor receptor (EGFR), a member of the tyrosine kinase family, plays an essential role in several cell signaling pathways, including angiogenesis, apoptosis, cell proliferation, and metastatic dissemination [40]. Additionally, EGFR activation may trigger other signaling pathways, all strongly linked to carcinogenesis, such as RAS/RAF/MEK, STAT, and PI3K/AKT/mTOR. As a consequence, the overexpression of EGFR has been linked to numerous cancers, including head, neck, lung, breast, colon, kidney, prostate, pancreas, brain, and ovarian cancer [40]. Considering the key role of EGFR in cancer development, its inhibitors were developed, with erlotinib and gefitinib representing the first-generation drugs, and osimertinib, cetuximab, afatinib, brigatinib, dacomitinib, panitumumab, and necitumumab the subsequent ones [40]. Ocular adverse effects, affecting especially the cornea, have been frequently observed with EGFR inhibitors, likely due to the crucial role of EGFR in the corneal wound healing process. In particular, the occurrence of corneal erosions and superficial punctate keratopathy may be linked to the inhibition of the EGFR role in wound healing [41,42].

The FDA-approved gefitinib is generally used to treat metastatic non-small-cell lung cancer. It was reported to determine ocular adverse effects in approximately 21% of cases in a study by Fukuoka et al. [43]. Specifically, conjunctivitis was reported with a prevalence of 1.4–14.5% of patients, without requiring treatment [44], while DED had an incidence rate of 4.1% to 8.1% [45].

Superficial punctate keratopathy occurred in 1.8% to 2.4% of cases, without the need for treatment discontinuation [45], while corneal erosions, reported at a rate of 1.2% to 1.8% of cases, could be more severe, with the potential need to temporarily or permanently interrupt the anticancer treatment [45,46]. Erlotinib, utilized for metastatic pancreatic cancer and non-small-cell lung cancer treatment, has been linked to corneal side effects as the most serious ocular surface adverse effects encountered by patients. Five cases of corneal ulcers were reported in the literature; most of them required the treatment to b stopped, in conjunction with antibiotics, and, in one case, penetrating keratoplasty was deemed necessary [47,48,49,50,51].

Moreover, keratitis, keratouveitis, and corneal edema were also reported during erlotinib treatment, but all of them were resolved after therapy discontinuation [52,53,54]. Diffuse punctate keratopathy was noted in two eyes, managed with moxifloxacin, tear substitutes, and a temporary discontinuation of treatment [55]. Corneal perforations occurred in two cases; one had to discontinue the treatment and underwent penetrating keratoplasty, while the other one was managed with a reduction in the dose of erlotinib [56,57].

Afatinib is another EGFR inhibitor approved for the treatment of non-small-cell lung cancer. The FDA estimated that 0.7% of patients may develop keratitis; for this reason, the FDA advises to delay the use of afatinib while keratitis is evaluated and to stop it if ulcerative keratitis is diagnosed [58]. In addition to the discontinuation of the medication, in the case of afatinib-related ulcerative keratitis, oral corticosteroids, doxycycline, ofloxacin, and corticosteroid drops may be needed to manage this adverse event [59]. Moreover, Afatinib patients could develop periorbital edema and conjunctivitis in 3.3% of cases [60].

Osimertinib has the same therapeutical indication of afatinib, especially for the metastatic forms. According to its prescribing information, only 0.7% of patients had keratitis and, in this instance, patients should be referred to the ophthalmologist as soon as possible for a complete assessment [61]. Rarely, osimertinib can cause vortex keratopathy; two cases have been documented in the literature, and a trial revealed an incidence of 0.5% [62,63], with patients showing bilateral moderate corneal deposits in a whorl pattern at the level of the basal epithelium [63]. The suppression of corneal epithelial cell migration, which relies on epidermal growth factor signaling, is thought to be the pivotal mechanism. Nevertheless, in both cases, patients were able to continue their anticancer treatment using topical lubricant agents in conjunction with osimertinib since the keratopathy seemed not to impair their vision [63].

Similar to the previous drugs, dacomitinib and necitumumab were also approved for the treatment of metastatic non-small-cell lung cancer. Both these anticancer therapies may determine conjunctivitis or keratitis, but neither of them requires interruption of treatment, since such adverse effects can be easily managed with topical therapy [64,65].

Cetuximab is recommended for the treatment of colorectal and head and neck cancers and has been reported to mainly cause eyelid rash and DED. However, these side effects can be easily managed through eyelid hygiene and tear substitute use [66]. Three cases of keratitis (one of them diagnosed as filamentary keratitis) have also been described, suggesting the possibility to use the topical epidermal growth factor as an effective treatment for this corneal complication [67,68,69].

Lastly, panitumumab, which is authorized for the treatment of metastatic colorectal cancer, has been associated with trichomegaly, similarly to cetuximab, afatinib, erlotinib, and gefitinib [70]. The FDA reported that trichomegaly could affect up to 6% of panitumumab users [70]. In addition, one case of corneal perforation was reported, and discontinuation of the medication along with penetrating keratoplasty was required for its management [50].

### 3.3. Fibroblast Growth Factor Receptor Inhibitors

A family of tyrosine kinases known as fibroblast growth factor receptors (FGFRs) has been linked to carcinogenesis, particularly in cases of lung adenocarcinoma, ovarian cancer, hepatocellular carcinoma, and urothelial carcinoma [71]. The two main FGFR inhibitors are ponatinib and erdafitinib. In particular, erdafitinib is a pan-FGFR inhibitor authorized for the treatment of metastatic urothelial cancer [72]. However, it is also being investigated for cholangiocarcinoma, liver cancer, non-small-cell lung cancer, prostate cancer, lymphoma, and esophageal cancer [71].

While no ocular surface adverse effects were found for ponatinib, erdafitinib was shown to cause corneal thinning and keratitis [73,74]. However, corneal thinning was found to be reversible with drug discontinuation [74], while keratitis was effectively treated with strict ophthalmological follow-up, topical autologous serum, and DED therapy [73]. As shown for EGFR inhibitors, FGFR inhibitors also appear to be responsible for ocular surface adverse effects due to the inhibition of corneal repair processes [75].

### 3.4. Human Epidermal Growth Factor 2 Inhibitors

A tumor-associated antigen called human epidermal growth factor receptor 2 (HER2) is overexpressed or detected in around 25% of patients with breast cancer [76]. Since their development, HER2 inhibitors have significantly increased median overall survival up to 57 months [76]. Lepatinib, neratinib, and tucatinib are examples of novel HER2 inhibitors that have been developed and added to the original HER2 inhibitors, trastuzumab and pertuzumab. Regarding ocular surface adverse effects, the FDA and literature have not reported any ocular side effects for lapatinib, neratinib, or tucatinib, while there have been recorded ophthalmic side effects for trastuzumab and pertuzumab.

The main documented ocular surface adverse events of trastuzumab are conjunctivitis, infectious crystalline keratopathy, and corneal ulceration. Conjunctivitis does not require any treatment modification, while corneal ulceration and infectious crystalline keratopathy can be treated with autologous serum and topical antibiotics, in addition to drug discontinuation [77,78,79].

On the other hand, pertuzumab is only associated with increased lacrimation, which was reported in three clinical trials [80]. For this class of drugs, it has been hypothesized their ocular side effects could be linked to either an impairment in corneal epithelial cells turnover or to an immunologically mediated mechanism of antibody-dependent cell cytotoxicity with consequent corneal damage [79].

### 3.5. Mitogen-Activated Protein Kinase Inhibitors

Mitogen-activated protein kinase (MEK) is a kinase protein part of the mitogen-activated protein kinase (MAPK) pathway (RAS-RAF-MEK-ERK), which regulates gene expression, cell cycle and proliferation, and can be overactivated in several cancers [81]. Precisely, this signaling pathway is highly involved in the development of BRAF-mutated melanoma, KRAS/BRAF-mutated colorectal cancer, and metastatic non-small-cell lung cancers [81]. MEK inhibitors aim to block MEK, thus interrupting this intracellular signaling cascade. To date, trametinib, cobimetinib, selumetinib, and binimetinib are four types of MEK inhibitors approved by the FDA for clinical use [81]. Several ocular adverse events were reported; among these, anterior segment disorders were quite common [13]. Specifically, DED after the use of trametinib, cobimetinib, and binimetinib was reported with an incidence between 2% and 10% [82]. A single case of bilateral corneal stromal opacity and epithelial microcystic edema was documented after cobimetinib use combined with vemurafenib and resolved with preservative-free tear substitutes. Failure of the corneal endothelium pump was suggested as the potential cause of the corneal decompensation reported in one case [83]. Furthermore, a phase I trial performed with the more selective MEK inhibitor RO4987655 showed signs of ocular toxicity in 27% of patients, who, among others, displayed punctate keratitis, corneal erosions, and DED [84].

### 3.6. BRAF Inhibitors

BRAF is a kinase involved in the MAPK pathway. Its mutation is found in the pathogenesis of several tumors [85]. To date, the FDA has approved three BRAF inhibitors: vemurafenib, dabrafenib, and encorafenib. These are indicated for treating BRAF mutation cancers, including unresectable or metastatic cutaneous melanoma, metastatic non-small-cell lung cancer, metastatic anaplastic, thyroid cancer, and metastatic colorectal cancer [86,87,88]. The main side effects involving the ocular surface after the use of vemurafenib were DED (2%), conjunctivitis (2.8%), and blepharitis (0.5%), all of which managed without suspending the drug [89,90].

DED was further associated with the treatment of both dabrafenib and encorafenib [90,91]. Notably, in a retrospective study, encorafenib, which represents the latest FDA-approved BRAF inhibitors, has been shown to induce ocular adverse events at an earlier treatment stage compared to other members of the same drug class. Nevertheless, these side effects were managed with local therapy, avoiding the suspension of anticancer therapy [91].

In addition, as mentioned previously, the use of vemurafenib in association with cobimetinib determined one case of corneal decompensation, that, however, resolved spontaneously without drug discontinuation [83]. To date, the exact mechanisms underlying these ocular adverse effects are still unknown, and further studies are needed to better understand their pathophysiology.

### 3.7. Selective Estrogen Receptor Modulator

Breast tissue differentiation and proliferation are mediated by female hormones, specifically estrogen and progesterone. The overexpression of estrogen receptors occurs in approximately 70% of breast cancer cases, motivating the role of endocrine therapy in breast cancer management [92].

Tamoxifen, raloxifene, and toremifene belong to the class of drugs known as selective estrogen receptor modulators (SERMs). By exerting tissue-selective agonist or antagonist properties in their interactions with estrogen receptors (ERs), they represent a milestone in breast cancer treatment [13].

Ocular surface physiology heavily depends on the balance between estrogen and androgen, and treatment-related hormone variations may determine ocular surface dysfunctions [93].

Several studies reported an association between tamoxifen and keratopathy. Different corneal changes were associated with the use of this drug, including subepithelial changes, causing vortex keratopathy and stromal opacities [94]. Based solely on clinical examination, the prevalence of keratopathy in tamoxifen users seems to not exceed 10% [94,95,96]. However, a confocal microscopy study found a prevalence of keratopathy above 70% [97]. Generally, cases of symptomless keratopathy can be managed with observation only. Nonetheless, temporary suspension or discontinuation of tamoxifen could be considered in symptomatic cases [97]. Furthermore, although tamoxifen keratopathy is thought to be reversible, the persistence of drug-induced corneal abnormalities after drug discontinuation has been described [98].

Raloxifene treatment is indicated not only for managing breast cancer but also for the prevention and treatment of osteoporosis. In this context, a case of verticillate cornea was reported, and the keratopathy did not improve at the 3-month follow-up after drug discontinuation. Despite the unknown pathogenetic mechanism, it was supposed that raloxifene might cause lipid accumulation through either the inhibition of lysosomal phospholipase or through the formation of a non-degradable drug–lipid complex [99].

Toremifene mainly showed anterior segment side effects. In particular, 9% of patients showed DED and 3% keratopathy [13,100].

### 3.8. Aromatase Inhibitors

As formerly reported, several types of breast cancers are sensitive to female hormones. In postmenopausal women, the production of estrogens within ovaries decreases and it is guaranteed by the presence of aromatase enzymes in peripheral tissues. Therefore, to lower estrogen levels, aromatase inhibitors represent a valid treatment option for postmenopausal women with hormone receptor-positive breast cancer. Anastrozole, letrozole, and exemestane are the three FDA-approved aromatase inhibitors, which have changed the story of hormone therapy for breast cancers [13]. As already explained, considering that the balance between estrogen and androgen plays a crucial role in the maintenance of a normal tear film and in the pathophysiology of ocular surface diseases, such as DED, blepharitis, and meibomian gland dysfunction, the onset of these disorders could be ascribable to hormone variations induced by endocrine therapy [93].

Chatziralli et al. reported a high prevalence of ocular surface diseases in patients treated with aromatase inhibitors. In detail, 75% developed blepharitis, 42.5% meibomian gland dysfunction, 30% superficial punctate keratitis, and 22.5% conjunctival injection [101]. In addition, DED was reported to occur in a range between 29% and 35% of patients treated with aromatase inhibitors [102,103]. Most cases were managed using conservative approaches, like tear substitutes, warm compresses, and others, without the need to stop anticancer therapy [13].

Additionally, considering each single aromatase inhibitor, a case of intraepithelial microcysts was reported after the use of exemestane, suggesting a possible drug-induced inhibition on limbal cell function [104], and a possible association between the onset of Sjogren syndrome and anastrozole/letrozole use was documented [105,106,107].

### 3.9. Breakpoint Cluster Region–Abelson Oncogene Inhibitors

The breakpoint cluster region–Abelson oncogene (BCR-ABL) is a hybrid gene known as Philadelphia chromosome that translates into a protein tyrosine kinase responsible for cell proliferation, differentiation, and apoptosis. Imatinib, the first tyrosine kinase inhibitor approved in 2001, drastically changed the treatment of chronic myeloid leukemia. Subsequently, second-generation and third-generation BCR-ABL inhibitors were developed, including nilotinib, dasatinib, bosutinib, and ponatinib [108].

Imatinib has been shown to determine several ocular side effects, with anterior segment toxicity being quite common, and some cases of conjunctivitis and keratoconjunctivitis were reported [13,109,110,111]. Nilotinib has been associated with the onset of DED. Specifically, in a phase II clinical trial evaluating the treatment of metastatic melanoma with nilotinib, DED represented the most common side effect, with 33.3% of enrolled patients affected by this complication [112]. Moreover, a case of DED that resolved after drug suspension was also reported in chronic myeloid leukemia. The suppression of c-kit receptors on the conjunctiva could potentially be responsible for ocular complications observed with the use of nilotinib [113]. Conversely, dasatinib, bosutinib, and ponatinib did not seem to determine any ocular surface adverse effects.

### 3.10. Fms-Like Tyrosine Kinase 3 Inhibitors

Fms-like tyrosine kinase 3 (FLT3) belongs to the class III receptor tyrosine kinase family and plays a crucial role in the proliferation of hematopoietic cells and lymphocytes. Since FLT3 mutations were found in about one-third of patients with acute myeloid leukemia, several targeted FLT3 inhibitors have been developed. Particularly, based on their potency and specificity, FLT3 inhibitors are divided into first-generation inhibitors, such as sunitinib, sorafenib, midostaurin, and ponatinib, and second-generation inhibitors, such as quizartinib, crenolanib, regorafenib, and gilteritinib [13].

Several cases of superficial ocular anterior toxicity and blurred vision were reported after the administration of sunitinib and sorafenib. These effects may be attributed to either the drug itself or to its metabolites, which may irritate and damage ocular structures such as the eyelids and the lacrimal gland [114,115]. Additionally, a 6.3% rate of DED was reported after the use of gilteritinib in a Japanese study [116].

### 3.11. Bruton’s Tyrosine Kinase Inhibitors

Bruton’s tyrosine kinase (BTK) is a tyrosine kinase, encoded by the BTK gene, which plays a crucial role in B-cell development, proliferation, and differentiation. BTK inhibitors represent alternative agents to chemotherapy-based regimens and are indicated in patients with B-cell malignancies but also disorders such as chronic graft-versus-host disease. FDA-approved BTK inhibitors are ibrutinib, acalabrutinib, and zanubrutinib [117].

Ibrutinib, the first one to be approved, has been linked to some ocular surface side effects. Byrd et al. reported that 10% of patients treated with ibrutinib developed unexplained blurred vision [118]. Moreover, another trial showed 6.5% of patients unveiled DED and ocular redness, which were successfully treated with topical therapy without the interruption of systemic treatment [119]. Nonetheless, as with BRAF inhibitors, no pathophysiological mechanism underlying these ocular side effects has yet been demonstrated. No ocular surface side effects were documented for acalabrutinib and zanubrutinib.

### 3.12. Anaplastic Lymphoma Kinase Inhibitors

Anaplastic lymphoma kinase (ALK) is a receptor tyrosine kinase discovered in anaplastic large-cell lymphoma cells. The ALK gene, after genic mutations, can become oncogenic. In this instance, cancers are considered to be ALK-positive. ALK inhibitors have shown significant activity in ALK-positive tumors, especially in non-small-cell lung cancer. Crizotinib was the first ALK inhibitor approved by the FDA, followed by ceritinib, alectinib, lorlatinib, and entrectinib [120].

A series of ocular side effects considered possible complications of non-ALK tyrosine kinase activation have been documented in patients treated with ALK inhibitors. Regarding the ocular surface, one case of persistent chronic conjunctival chemosis not improving despite treatment with oral acetazolamide and corticosteroids was described with crizotinib. Nonetheless, after drug suspension, conjunctival chemosis resolved in one week, showing a possible direct inflammatory effect of this drug on the conjunctiva [121]. In a phase I trial, the combination therapy of crizotinib and erlotinib, an EGFR inhibitor, caused DED in some patients treated for advanced non-small-cell lung cancer. However, in such cases, it was not possible to determine which drug contributed the most to the DED onset [122]. No other ocular surface side effects were reported.

### 3.13. Vascular Endothelial Growth Factor Receptor Inhibitors

The vascular endothelial growth factor (VEGF) is a strong pro-angiogenic factor implicated in the development of both physiological vascular networks and pathological ones in conditions, such as tumor growth, metastasis, and ocular neovascular disorders. Indeed, to limit the pathological pathway activation in several cancers, some VEGF and VEGF receptor (VEGFR) inhibitors have been developed, including sorafenib, sunitinib, vandetanib, lenvatinib, axitinib, bevacizumab, ramucirumab, cabozantinib, and pazopanib [13].

Pazopanib, an orally administered VEGFR inhibitor approved for renal cell carcinoma and soft tissue sarcoma that inhibits all VEGF receptors subtypes, was found to cause signs and symptoms of anterior segment toxicity, comprising ocular irritation, dry eyes, keratitis, and chemosis [115]. Moreover, when evaluated as eye drops to treat patients with neovascular age-related macular degeneration, its monotherapy did not improve macular disease, and the most common ocular side effect was mild ocular irritation [123].

Vandetanib is a kinase inhibitor, which targets both EGFR and VEGFR and is used for medullary thyroid cancer and non-small-cell lung cancer. The FDA reported corneal alterations possibly altering visual acuity in patients on vandetanib therapy and recommended ophthalmological evaluation in patients with reported visual changes [124]. Vortex keratopathy, characterized by a whorl-like pattern, was documented in different reports. In particular, Ahn et al. described a case of vortex keratopathy after six months of vandetanib [125]. The same disease occurred in a patient on vandetanib therapy for anaplastic astrocytoma, which was managed with dose reduction and lubrication with carboxymethylcellulose [126]. Furthermore, Shin et al. described more than 15% of patients under vandetanib treatment developing vortex keratopathy that resolved with drug discontinuation [62].

Although the pathological mechanism underlining such corneal disorders remains unclear, it has been hypothesized that either the accumulation of vandetanib metabolites in the corneal epithelium or a direct inhibition of the ocular EGFR may result in alterations of epithelial cell proliferation and migration [62,125]. No ocular surface side effects have been reported for all the other VEGFR inhibitors.

### 3.14. Proteasome Inhibitors

The ubiquitin proteasome system is an intracellular selective system that is crucial for the degradation of unfolded or damaged proteins. It is essential for many cellular processes, including cell cycle and apoptosis. Proteasome inhibitors have anti-tumor activity by inactivating pro-growth proteins and blocking cell proliferation. Bortezomib, carfilzomib, and ixazomib are the FDA-approved proteasome inhibitors [13].

Bortezomib, a first-generation proteasome inhibitor, is used in the treatment of multiple myeloma, mantle cell lymphoma, and other hematologic malignancies. Several cases of chalazion and blepharitis were reported after its use [111]. Despite uncertainties about the pathogenesis, an inflammatory process linked to the accumulation of degraded proteins and the release of pro-inflammatory cytokines in the eyelid is considered plausible. In support of this hypothesis, the literature has highlighted a good response to anti-inflammatory therapy, even though in some serious cases, the suspension of the anticancer treatment was needed [127,128,129]. Sklar et al. also reported the use of carfilzomib, a second-generation proteasome inhibitor, associated with the development of chalazia and blepharitis in two patients previously treated with bortezomib [127].

Lastly, ixazomib, approved for the treatment of multiple myeloma, determined different ocular surface adverse events, with blurred vision and conjunctivitis being the most common at a rate of 6%, followed by DED at a rate of 5%.

### 3.15. Antibody–Drug Conjugates

A monoclonal antibody and a lethal payload targeted to a particular antigen on a malignant cell are combined in a unique way known as antibody–drug conjugates [130]. This approach can spare healthy cells while delivering highly lethal medications to targeted tumor cells [126]. Trastuzumab deruxtecan, belantamab mafodotin, trastuzumab emtansine, enfortumab vedotin, gemtuzumab orogamicin, inotuzumab ozogamicin, sacituzumab govitecan, and mirvetuximab soravtansine are among the antibody–drug conjugates that have received FDA approval.

Adverse effects on the ocular surface of this class of drugs are supposed to be determined by the combination of side effects of the individual conjugated drugs, with a possible enhancement due to a synergistic mechanism [130].

Trastuzumab deruxtecan is an HER2-directed antibody and topoisomerase inhibitor prescribed to treat metastatic HER2-positive breast cancer [131]. According to data from the FDA, 11% of patients receiving trastuzumab deruxtecan reported DED, for which no change in therapy was needed.

The FDA also authorized belantamab mafodotin, a combination of microtubule inhibitors and antibodies directed against B-cell maturation antigens for the treatment of refractory multiple myeloma. Patients using this medication were shown to potentially develop corneal epithelial alterations resembling microcysts, which needed dose adjustments, reduction, or stop [132,133,134]. Before starting belantamab mafodotin and before each subsequent dose, the FDA advises to have a complete ophthalmological examination [135]. Moreover, the FDA suggests avoiding contact lenses use, to apply cold compresses for symptom alleviation, and to use preservative-free tear substitutes at least four times a day [135]. Additionally, belantamab mafodotin determines DED in 15% of patients and blurred vision in 25–34% of patients, with dose reductions being necessary in both these circumstances [136].

Trastuzumab emtansine, an HER2-targed antibody with a microtubule inhibitor, was authorized to treat metastatic breast cancer, and it may cause corneal epithelium abnormalities, mainly related to the effects of trastuzumab. Hyaluronic acid, autologous serum, or observation are all possible options to manage these ocular adverse events, without the need to change anticancer therapy and dosage [137,138,139].

Enfortumab vedotin, a nectin-4 targeting antibody with a microtubule inhibitor, was FDA approved to treat metastatic urothelial carcinoma [140]. The FDA reported that 36–40% of patients on enfortumab vedotin had DED and 14% had impaired vision [140]. Therefore, routine monitoring for ocular disorders was suggested, as well as tear substitute use for DED prophylaxis and topical corticosteroids if needed [140].

Mirvetuximab soravtansine was approved as a first-in-class antibody–drug conjugate targeting folate receptor alpha for the treatment of adult patients with folate receptor alpha-positive, platinum-resistant epithelial ovarian, fallopian tube, or primary peritoneal cancer who previously underwent one to three systemic treatment regimens [141].

Reversible corneal epithelium alterations were reported, including keratopathy and DED, likely related to “off-target” effects of this drug, which may exert antimitotic effects on dividing cells within the corneal epithelium, leading to disruption of the epithelial layer [142,143]. No reports of corneal ulcers or perforations were found among the mirvetuximab soravtansine-associated ocular surface adverse events. Ocular surface-related side effect prevention strategies are based on recommendations for maintaining ocular surface health, such as daily use of lubricating eye drops and occasional use of corticosteroid eye drops. Eye examination should be performed at baseline, every other cycle for the first eight cycles of treatment, and dose modifications should also be considered if clinically indicated [141].

No ocular surface side effects have been reported for sacituzumab govitecan, inotuzumab ozogamicin, and gemtuzumab orogamicin.

## 4. Mechanism of Drugs Side Effects

Focusing on the pathophysiological mechanisms underlying the ocular surface side effects of the novel discussed anticancer drugs, four different main mechanisms have been identified.

The first mechanism is represented by the interference with the normal processes of corneal wound healing and repair and by the impairment in corneal epithelial cell turnover. This mechanism appears to be linked to the corneal damage underlying anticancer drugs, such as EGFR inhibitors [41,42], FGFR inhibitors [75], HER2 inhibitors [79], and, presumably, vandetanib, one of the VEGFR inhibitors [62,125].

Furthermore, for some drugs, such as MEK inhibitors and BRAF inhibitors, a mechanism of corneal decompensation has also been hypothesized due to failure of the corneal endothelium pump [83]. Although not being a side effect directed towards the ocular surface, the damage to the inner corneal layer can subsequently lead to corneal epithelial alterations. Another possible pathophysiological mechanism that seems to characterize some of the new anticancer drugs is represented by immunological mechanisms, determining the accumulation of antibodies or degradation metabolites. This causes a cell cytotoxicity effect that seems to be responsible, in particular, for DED and corneal damage. This mechanism appears to characterize the ocular surface side effects of immune checkpoint inhibitors [12,13], HER2 inhibitors [79], BCR-ABL inhibitors [112,113], and proteasome inhibitors, particularly bortezomib [127,128,129].

Finally, a last possible mechanism of action underlying the ocular surface side effects is represented by hormonal imbalance, which is known to determine tear film instability and dysfunction of some gland structures, leading to DED, blepharitis, and meibomian gland dysfunction. In particular, the side effects of SERMs and aromatase inhibitors appear to be attributable to this pathophysiological mechanism [93].

However, these are only some of the potential pathophysiological mechanisms that have been identified or hypothesized, but for other classes of drugs, such as ALK inhibitors, BTK inhibitors, and FLT3 inhibitors, the cause of the described side effects still remains unknown, and further research is needed to better understand the physiopathological basis of such adverse effects on the ocular surface.

In comparison with traditional anticancer drugs, some differences in the incidence of ocular surface side effects and their pathophysiological mechanisms can be noticed. In fact, traditional drugs showed higher incidence of ocular surface side effects compared to the more recent ones, which are better tolerated, while the main pathophysiological mechanism is represented by the powerful cytotoxic effect of the traditional drugs. On the contrary, the type of side effects and their management appear to be similar, with less severity in the case of novel anticancer drugs.

A detailed description of the types and management of ocular surface adverse effects caused by traditional and novel anticancer drugs is summarized in Table 1 and Table 2, respectively.

## 5. Discussion

The development of new, innovative, and targeted anticancer drugs has markedly changed the methods of treating cancer, improving the prognosis of affected patients. However, anticancer drugs are not devoid of treatment-related adverse events that can involve the eyes. Despite novel anticancer drugs being seemingly better tolerated compared to traditional and conventional chemotherapy, ocular side effects affecting the ocular surface can still occur.

The vast majority of novel anticancer drug-associated ocular surface side effects can be easily managed with observation, topical treatments, and, at most, a reduction in the anticancer drug dosage. Only in a minority of cases, drug discontinuation may be required.

The most common ocular surface adverse effects are DED, conjunctivitis, and corneal damage, while blepharitis and meibomian gland dysfunction appear to be related to only a few of these new drugs, such as immune checkpoint inhibitors, BRAF inhibitors, aromatase inhibitors, and proteasome inhibitors.

Treatment of the ocular surface adverse effects varies according to clinical signs and symptoms.

Dry eye disease should be managed following the guidelines provided by the Tear Film Ocular Surface Dry Eye Workshop II (TFOS DEWS II), which currently represent the most accredited and valid source for DED management [93].

Conjunctivitis usually does not require treatment and observation and is generally performed given their tendency to quickly resolve quickly on their own. Nonetheless, careful slit lamp examination to exclude infectious and allergic etiologies could be necessary.

Cases of corneal damage, such as keratitis, vortex keratopathy, and corneal ulcers, represent the most serious adverse effect, which may require treatment suspension and, in the worst-case scenario, corneal transplant. For this reason, patients treated with drugs more frequently associated with severe corneal adverse events, such as immune checkpoint inhibitors, EGFR inhibitors, FGFR inhibitors, and HER2 inhibitors, should receive close ophthalmological examinations and should be referred to the eye specialist for an unscheduled visit in the case of the onset of ocular symptoms.

Lastly, ocular adnexal side effects, such as blepharitis and meibomian gland dysfunction, can be easily managed with topical therapy, including ocular lubricants, eyelid scrubs, and warm compresses, reserving systemic antibiotics only for severe cases.

Concerning the pathophysiological mechanisms underlying these ocular surface side effects, four main potential mechanisms have been identified or hypothesized. Specifically, interference with the physiological processes of corneal wound healing and repair is associated with impairment in corneal epithelial cell turnover, mainly leading to keratitis and corneal damage; corneal decompensation due to failure of the corneal endothelium pump; immunologically mediated mechanism of cell cytotoxicity due to antibodies or degradation product accumulation, mainly causing DED, meibomian gland dysfunction, blepharitis, and corneal alterations; and hormonal dysregulation, principally determining DED.

A central issue in drug risk–benefit assessment is to identify the frequencies and severity of side effects. In this context, a future pivotal challenge would be to identify predictive factors for ocular surface adverse events in order to either direct the patient towards a lower anticancer drug dosage, if possible, or alternative anticancer treatment, or to provide closer ophthalmological follow-up when needed. To date, no studies have focused on the detection of such predictors, and the only option is carry out a complete ophthalmological examination in patients selected for a new anticancer therapy to try to identify any risk factor or clinical sign of DED, blepharitis, meibomian gland dysfunction, or corneal alterations.

## 6. Conclusions

In conclusion, in order to guarantee proper management and to reduce potentially crippling ocular surface complications, both oncologists and ophthalmologists should be informed about the prevalence and the characteristics of the possible ocular adverse events occurring in patients receiving anticancer medications, especially for newer drugs that are able to provide better cancer survival and that showed to be better tolerated by the ocular surface than traditional anticancer drugs. Although pathological mechanisms have been identified or, in some instances, hypothesized, larger case–control studies are required to further elucidate the impact of ocular surface side effects on the clinical course of the oncological patient and hopefully to identify potential predictors for these occurrences.

## Figures and Tables

**Table 1 cancers-16-00344-t001:** Types and management of ocular surface adverse effects caused by traditional anticancer drugs.

Agent	Malignancy	Ocular Surface Adverse Effects	Management
Carboplatin	Lung cancer Breast cancer Renal cancer Ovarian cancer Head and neck cancers	Conjunctival injection Subconjunctival hemorrhage Corneal edema	Resolve quickly on its own. Careful slit lamp examination to exclude infectious and allergic etiologies -
Oxaliplatin	Colon cancer Esophageal cancer Pancreatic cancer Breast cancer Ovarian cancer	Conjunctivitis	Resolve quickly on its own. Careful slit lamp examination to exclude infectious and allergic etiologies
Chlorambucil	Leukemia Non-Hodgkin lymphoma Hodgkin’s disease	Keratitis	Ocular lubricant, topical antibiotic, bandage contact lens, considering discontinuing treatment or reducing dosage
Cyclophosphamide	Breast cancer Ovarian cancer Burkitt lymphoma Multiple myeloma	Blepharoconjunctivitis Corneal ulcer	Ocular lubricant, topical antibiotic, bandage contact lens, considering discontinuing treatment or reducing dosage
Ifosphamide	Lung cancer Breast cancer Ovarian cancer Testicular cancer	Conjunctivitis	Resolve quickly on its own. Careful slit lamp examination to exclude infectious and allergic etiologies
Busulfan	Leukemia	Keratoconjunctivitis sicca	Ocular lubricant, topical antibiotic, bandage contact lens, considering discontinuing treatment or reducing dosage
Cytarabine	Leukemia Lymphoma	Conjunctival hyperemia Conjunctival inflammation Superficial punctate keratitis Dry eye	Resolve quickly on its own. Careful slit lamp examination to exclude infectious and allergic etiologies. Prescribe 2-deoxycitidine as preventive care Ocular lubricant, topical antibiotic, bandage contact lens, considering discontinuing treatment or reducing dosage Manage according to the Tear Film Ocular Surface Dry Eye Workshop II guidelines
5-Fluorouracil	Breast cancer Skin cancer Head and neck cancers Renal cancer Cervical cancer Gastrointestinal cancers	Ocular irritation	Artificial tears for symptomatic relief
Capecitabine	Metastatic breast cancer Metastatic colon cancer Advanced gastric carcinoma	Corneal deposits Ocular irritation	Non-preserved artificial tears
Methotrexate	Breast cancer Head and neck cancers Leukemia Lymphoma	Dry eye	Manage according to the Tear Film Ocular Surface Dry Eye Workshop II guidelines. Prescribe folinic acid/leucovorin as preventive care
Pentostatin	Lymphoma Chronic lymphocytic leukemia	Conjunctivitis	Resolve quickly on its own. Careful slit lamp examination to exclude infectious and allergic etiologies
Docetaxel	Head and neck cancers Breast cancer Prostate cancer Gastric cancer	Erosive conjunctivitis	Erythromicin and considering reviewing chemotherapy treatment
Vincristine Vinblastine Vindesine Vinorelbine	Non-Hodgkin lymphoma Hodgkin’s disease Leukemia Ewing’s sarcoma Lung cancer Breast cancer	Corneal hypoesthesia	-
Doxorubicin	Lung cancer Breast cancer Ovarian cancer Sarcoma Acute leukemia Non-Hodgkin lymphoma Hodgkin’s disease	Conjunctivitis	Resolve quickly on its own. Careful slit lamp examination to exclude infectious and allergic etiologies
Combination Regimens			
Cyclophosphamide Methotrexate 5-Fluorouracil	Breast cancer	Conjunctivitis	Resolve quickly on its own. Careful slit lamp examination to exclude infectious and allergic etiologies
5-Fluorouracil Epirubicin Cyclophosphamide	Breast cancer	Blepharitis Conjunctival hyperemia Punctate epithelial keratopathy	Eyelid scrubs, warm compressions, systemic antibiotics for severe cases Resolve quickly on its own. Careful slit lamp examination to exclude infectious and allergic etiologies Ocular lubricant, topical antibiotic, bandage contact lens, considering discontinuing treatment or reducing dosage
Tegafur Gimeracil Oteracil	Lung cancer Gastric cancer Colon cancer Bile duct cancer Pancreatic cancer	Meibomian gland disease Corneal epithelium damage	Eyelid scrubs, warm compressions, systemic antibiotics for severe cases Ocular lubricant, topical antibiotic, bandage contact lens, considering discontinuing treatment or reducing dosage

**Table 2 cancers-16-00344-t002:** Types and management of ocular surface adverse effects caused by novel anticancer drugs.

**Agent**	**Malignancy**	**Ocular Surface Adverse Effects**	**Management**
Immune checkpoint inhibitors			
Ipilimumab Avelumab Durvalumab Nivolumab Pembrolizumab Atezolizumab Cemiplimab	Hodgkin lymphoma Colorectal cancer Gastric cancer Urothelial carcinoma Non-small-cell lung cancer Hepatocellular carcinoma Head and neck cancers	Dry eye Blepharitis Conjunctivitis Ulcerative keratitis Cicatrizing conjunctivitis Corneal transplant rejection Corneal perforation	Topical cyclosporine, artificial tears, punctal occlusion Eyelid scrubs, warm compressions, systemic antibiotics for severe cases Resolve quickly on its own. Careful slit lamp examination to exclude infectious and allergic etiologies Bandage contact lenses, corneal glue, topical autologous serum and optimizing the ocular surface with artificial tears and punctal occlusion. Depending on the severity and the patient’s response to ocular therapy, stopping anticancer therapy should be considered, as well as surgery (penetrating keratoplasty)
Epidermal Growth Factor Receptor Inhibitors			
Gefitinib Erlotinib Afatinib Osimertinib Dacomitinib Necitumumab Cetuximab Panitumumab	Non-small-cell lung cancer Pancreatic cancer Colorectal cancer Head and neck cancers	Conjunctivitis Dry eye Superficial punctate keratopathy Corneal ulcer Keratitis Keratouveitis Corneal edema Vortex keratopathy Eyelid rash Trichomegaly	Resolve quickly on its own. Careful slit lamp examination to exclude infectious and allergic etiologies Manage according to the Tear Film Ocular Surface Dry Eye Workshop II guidelines Ocular lubricant, topical antibiotic, bandage contact lens, considering discontinuing treatment or reducing dosage Eyelid hygiene
Fibroblast Growth Factor Receptor Inhibitors			
Erdafitinib	Urothelial cancer	Keratitis Corneal thinning	Ocular lubricant and topical autologous serum, considering discontinuing treatment or reducing dosage
Human Epidermal Growth Factor 2 Inhibitors			
Trastuzumab Pertuzumab	Breast cancer	Conjunctivitis Infectious crystalline keratopathy Corneal ulcer Increased lacrimation	Resolve quickly on its own. Careful slit lamp examination to exclude infectious and allergic etiologies. Autologous serum and topical antibiotics, in addition to drug discontinuation.
Mitogen-activated protein kinase inhibitors			
Trametinib Cobinetimib Binimetinib	Melanoma Colorectal cancer Non-small cell lung cancer	Dry eye	Manage according to the Tear Film Ocular Surface Dry Eye Workshop II guidelines
BRAF Inhibitors			
Vemurafenib Dabrafenib Encorafenib	Melanoma Non-small cell lung cancer Thyroid cancer Colorectal cancer	Dry eye Conjunctivitis Blepharitis	Manage according to the Tear Film Ocular Surface Dry Eye Workshop II guidelines Resolve quickly on its own. Careful slit lamp examination to exclude infectious and allergic etiologies Eyelid scrubs, warm compressions, systemic antibiotics for severe cases
Selective Estrogen Receptor Modulator			
Tamoxifen Raloxifene Toremifene	Breast cancer	Keratopathy Dry eye	Only observation. Temporary drug suspension or discontinuation could be considered in symptomatic cases Manage according to the Tear Film Ocular Surface Dry Eye Workshop II guidelines
Aromatase inhibitors			
Anastrozole Letrozole Exemestane	Breast cancer	Blepharitis Meibomian gland dysfunction Dry eye Superficial punctate keratitis Conjunctival injection	Eyelid scrubs, warm compressions, systemic antibiotics for severe cases Manage according to the Tear Film Ocular Surface Dry Eye Workshop II guidelines Only observation
Breakpoint cluster region–Abelson oncogene inhibitors			
Imatinib Nilotinib	Chronic myeloid leukemia	Conjunctivitis Keratoconjunctivitis Dry eye	Resolve quickly on its own or with the use of topical therapy. Careful slit lamp examination to exclude infectious and allergic etiologies Manage according to the Tear Film Ocular Surface Dry Eye Workshop II guidelines
Fms-like tyrosine kinase 3 inhibitors			
Gilteritinib	Acute myeloid leukemia	Dry eye	Manage according to the Tear Film Ocular Surface Dry Eye Workshop II guidelines
Bruton’s tyrosine kinase inhibitors			
Ibrutinib	B-cell malignancies	Dry eye	Manage according to the Tear Film Ocular Surface Dry Eye Workshop II guidelines
Anaplastic lymphoma kinase inhibitors			
Crizotinib	Non-small cell lung cancer	Conjunctival chemosis	Drug discontinuation
Vascular endothelial growth factor receptor inhibitors			
Pazopanib Vandetanib	Renal cancer Sarcoma Non-small cell lung cancer Thyroid cancer	Dry eye Keratitis Vortex keratopathy	Manage according to the Tear Film Ocular Surface Dry Eye Workshop II guidelines Drug discontinuation
Proteasome inhibitors			
Bortezomib Carfilzomib Ixazomib	Multiple myeloma	Chalazia Blepharitis Conjunctivitis Dry eye	Eyelid scrubs, warm compressions, systemic antibiotics for severe cases Resolve quickly on its own. Careful slit lamp examination to exclude infectious and allergic etiologies Manage according to the Tear Film Ocular Surface Dry Eye Workshop II guidelines
Antibody–Drug Conjugates			
Trastuzumab deruxtecan Belantamab mafodotin Trastuzumab emtansine Enfortumab vedotin Mirvetuximab soravtansine	Breast cancer Multiple myeloma Breast cancer Urothelial cancer Ovarian cancer	Dry eye Dry eye Keratopathy Dry eye Dry eye Keratopathy	Manage according to the Tear Film Ocular Surface Dry Eye Workshop II guidelines Avoiding contact lenses, applying cold compressions for symptomatic alleviation, and using preservative-free artificial tears at least four times a day Hyaluronic acid, autologous serum, or observation are all possible, without changing anticancer therapy and dosage Artificial tears and topical corticosteroids if necessary Using lubricating eye drops every day and occasionally using corticosteroid eye drops. In addition, patients should also have an eye examination at baseline, every other cycle for the first eight cycles of treatment, and dose modifications should also be considered if clinically indicated

## Data Availability

The data can be shared up on request.

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
