# Peer review of "Ocular Surface Side Effects of Novel Anticancer Drugs"

_cancers, 2024, doi:10.3390/cancers16020344_

Round 1

Reviewer 1 Report

Comments and Suggestions for Authors

The topic of the manuscript is meaningful, but some suggestions:

1. The author aims to explore the potential mechanism for ocular surface side effects of these novel drugs but less words about the mechanism of the side efffects related to these novel drugs. If similar mechanism exists for different novel drugs, it is better to summarize these novel drugs into on group.

2. The discussion just talks about the management, why no mechanism?

3. Too much references and the most new articles is less in the reference list.

Author Response

The topic of the manuscript is meaningful, but some suggestions:

1. The author aims to explore the potential mechanism for ocular surface side effects of these novel drugs but less words about the mechanism of the side effects related to these novel drugs. If similar mechanism exists for different novel drugs, it is better to summarize these novel drugs into on group.

RE: Thank you for your comments and suggestions. We added a new paragraph (4. Mechanism of Drugs Side Effects) where we summarized the main potential mechanisms for ocular surface side effects of these novel anticancer drugs, also grouping drugs with similar pathophysiological mechanism (pages 11-12, lines 476-551).

“4. Mechanism of Drugs Side Effects

Focusing on the pathophysiological mechanisms underlying the ocular surface side effects of the novel discussed anticancer drugs, four different main mechanisms have been identified.

The first mechanism is represented by the interference with the normal processes of corneal wound healing and repair and by the impairment in corneal epithelial cell turnover. This mechanism appears to be linked to the corneal damage underlying anticancer drugs such as EGFR inhibitors [41,42], FGFR inhibitors [75], HER2 inhibitors [79] and, presumably, vandetanib, one of the VEGFR inhibitors [62,132].

Furthermore, for some drugs such as MEK inhibitors and BRAF inhibitors, a mechanism of corneal decompensation has also been hypothesized due to failure of the corneal endothelium pump [87]. Although not being a side effect directed towards the ocular surface, the damage to the inner corneal layer can subsequently lead to corneal epithelial alterations. Another possible pathophysiological mechanism that seems to characterize some of the new anticancer drugs is represented by immunological mechanisms determining accumulation of antibodies or degradation metabolites. This causes a cell cytotoxicity effect, that seems to be responsible in particular for DED and corneal damage. This mechanism appears to characterize the ocular surface side effects of immune checkpoint inhibitors [12,13], HER2 inhibitors [79], BCR-ABL inhibitors [119,120] and proteasome inhibitors, particularly Bortezomib [135-137].

Finally, a last possible mechanism of action underlying the ocular surface side effects is represented by hormonal imbalance, which is known to determine tear film instability and dysfunction of some gland structures, leading to DED, blepharitis and meibomian gland dysfunction. In particular, side effects of SERMs and aromatase inhibitors appear to be attributable to this pathophysiological mechanism [97].

However, these are only some of the potential pathophysiological mechanisms that have been identified or hypotezised, but for other classes of drugs such as ALK inhibitors, BTK inhibitors and FLT3 inhibitors, the cause of the described side effects still remains un-known and further research is needed to better understand the physiopathological basis of such adverse effects on the ocular surface.

In comparison with traditional anticancer drugs, some differences in the incidence of ocular surface side effects and their pathophysiological mechanisms can be noticed. In fact, traditional drugs showed higher incidence of ocular surface side effects compared to the more recent ones, which are better tolerated, while the main pathophysiological mechanism is represented by the powerful cytotoxic effect of the traditional drugs. On the contrary, the type of side effects and their management appear to be similar, with less severity in the case of novel anticancer drugs.

Detailed description of types and management of ocular surface adverse effects caused by traditional and novel anticancer drugs are summarized in tables 1 and 2, respectively.”

2. The discussion just talks about the management, why no mechanism?

RE: Thank you for your comment. We modified the Discussion section as you suggested (pages 19-20, lines 593-603).

“Concerning the pathophysiological mechanisms underlying these ocular surface side effects, four main potential mechanisms have been identified or hypothesized. Specifically, interference with the physiological processes of corneal wound healing and repair associated with impairment in corneal epithelial cell turnover mainly leading to keratitis and corneal damage; corneal decompensation due to failure of the corneal endothelium pump; immunologically mediated mechanism of cell cytotoxicity due to antibodies or degradation products accumulation mainly causing DED, meibomian gland dysfunction, blepharitis and corneal alterations; and hormonal dysregulation, principally determining DED.”

3. Too much references and the most new articles is less in the reference list.

RE: thank you for your suggestion. We modified and updated the reference list.

Reviewer 2 Report

Comments and Suggestions for Authors

see attachment

Author Response

Manuscript “Ocular Surface Side Effects of Novel Anticancer Drugs“ by Livio Vitiello, Filippo Lixi, Giulia Coco, Giuseppe Giannaccare is well written review on side effect of anticancer drugs which covers almost 40 years of research. It is quite timely contribution which difently fits to scope of Cancers.

RE: Thanks for your positive comments.

Generally manuscript gives very nice impression where authors grouped novel drugs in 15 different groups and discussed possible side effects on eyes and additional treatment required to overcome these problems. It is quite surprising that discussion is almost table single Table and the main conclusion is that „mechanism and prevalence of ocular side effects … are not fully understood yet“. On my mind authors should elaborate discussion much more:

1. Already known points for pathological mechanism should be mentioned and discussed.

RE: Thank you for your comments and suggestions. We added a new paragraph (4. Mechanism of Drugs Side Effects) where we summarized the main potential mechanisms for ocular surface side effects of these novel anticancer drugs, also grouping drugs with similar pathophysiological mechanism (pages 11-12, lines 476-551).

“4. Mechanism of Drugs Side Effects

Focusing on the pathophysiological mechanisms underlying the ocular surface side effects of the novel discussed anticancer drugs, four different main mechanisms have been identified.

The first mechanism is represented by the interference with the normal processes of corneal wound healing and repair and by the impairment in corneal epithelial cell turnover. This mechanism appears to be linked to the corneal damage underlying anticancer drugs such as EGFR inhibitors [41,42], FGFR inhibitors [75], HER2 inhibitors [79] and, presumably, vandetanib, one of the VEGFR inhibitors [62,132].

Furthermore, for some drugs such as MEK inhibitors and BRAF inhibitors, a mechanism of corneal decompensation has also been hypothesized due to failure of the corneal endothelium pump [87]. Although not being a side effect directed towards the ocular surface, the damage to the inner corneal layer can subsequently lead to corneal epithelial alterations. Another possible pathophysiological mechanism that seems to characterize some of the new anticancer drugs is represented by immunological mechanisms determining accumulation of antibodies or degradation metabolites. This causes a cell cytotoxicity effect, that seems to be responsible in particular for DED and corneal damage. This mechanism appears to characterize the ocular surface side effects of immune checkpoint inhibitors [12,13], HER2 inhibitors [79], BCR-ABL inhibitors [119,120] and proteasome inhibitors, particularly Bortezomib [135-137].

Finally, a last possible mechanism of action underlying the ocular surface side effects is represented by hormonal imbalance, which is known to determine tear film instability and dysfunction of some gland structures, leading to DED, blepharitis and meibomian gland dysfunction. In particular, side effects of SERMs and aromatase inhibitors appear to be attributable to this pathophysiological mechanism [97].

However, these are only some of the potential pathophysiological mechanisms that have been identified or hypotezised, but for other classes of drugs such as ALK inhibitors, BTK inhibitors and FLT3 inhibitors, the cause of the described side effects still remains un-known and further research is needed to better understand the physiopathological basis of such adverse effects on the ocular surface.

In comparison with traditional anticancer drugs, some differences in the incidence of ocular surface side effects and their pathophysiological mechanisms can be noticed. In fact, traditional drugs showed higher incidence of ocular surface side effects compared to the more recent ones, which are better tolerated, while the main pathophysiological mechanism is represented by the powerful cytotoxic effect of the traditional drugs. On the contrary, the type of side effects and their management appear to be similar, with less severity in the case of novel anticancer drugs.

Detailed description of types and management of ocular surface adverse effects caused by traditional and novel anticancer drugs are summarized in tables 1 and 2, respectively.”

2. What about predictions for side effects?

RE: Thank you for your precious comment. We modified the Discussion section according to your suggestions (page 20, lines 604-620).

“A central issue in drug risk-benefit assessment is to identify frequencies and severity of side effects. In this context, a future pivotal challenge would be to identify predictive factors for ocular surface adverse events in order to either direct the patient towards lower anticancer drug dosage if possible or alternative anticancer treatment, or to provide closer ophthalmological follow-up when needed. To date, no studies have focused on the detection of such predictors, and the only option is carry out a complete ophthalmological examination in patients selected for a new anticancer therapy to try to identify any risk factor or clinical sign of DED, blepharitis, meibomian gland dysfunction or corneal alterations. In conclusion, in order to guarantee proper management and to reduce potentially crippling ocular surface complications, both oncologists and ophthalmologists should be informed about the prevalence and the characteristics of the possible ocular adverse events occurring in patients receiving anticancer medications, especially for newer drugs that are able to provide better cancer survival. Although pathological mechanisms have been identified or in some instances hypothesized, larger case-controlled studies are required to further elucidate the impact of ocular surface side effects on the clinical course of the oncological patient and hopefully to identify potential predictors for these occurrences.”

3. Comparison between traditional and novel drugs from point of view ocular surface side effects and their treatment.

RE: thank you for your suggestion. We added this comparison between traditional and novel drugs (page 12, lines 543-551).

“In comparison with traditional anticancer drugs, some differences in the incidence of ocular surface side effects and their pathophysiological mechanisms can be noticed. In fact, traditional drugs showed higher incidence of ocular surface side effects compared to the more recent ones, which are better tolerated, while the main pathophysiological mechanism is represented by the powerful cytotoxic effect of the traditional drugs. On the contrary, the type of side effects and their management appear to be similar, with less severity in the case of novel anticancer drugs.

Detailed description of types and management of ocular surface adverse effects caused by traditional and novel anticancer drugs are summarized in tables 1 and 2, respectively.”

4. List of literature should be carefully checked, some names are not properly written.

RE: thank you for your suggestion. We modified, reviewed and updated the reference list.

Round 2

Reviewer 1 Report

Comments and Suggestions for Authors

no

Author Response

Comments and Suggestions for Authors: No

RE: thank you very much for your precious comments and suggestions that permitted to improve our manuscript.

Reviewer 2 Report

Comments and Suggestions for Authors

Authors have made significant improvements of manuscript and added new parts of discussion. One small thing i.e, conclusion is removed. Is it correct? I am sure that authors should polish text more but I hope it could be done with technical editor of Cancers

Author Response

Authors have made significant improvements of manuscript and added new parts of discussion. One small thing i.e, conclusion is removed. Is it correct? I am sure that authors should polish text more but I hope it could be done with technical editor of Cancers.

RE: Thank you  very much for your comment. We added the Conclusion section at the end of our manuscript (page 20).

In conclusion, in order to guarantee proper management and to reduce potentially crippling ocular surface complications, both oncologists and ophthalmologists should be informed about the prevalence and the characteristics of the possible ocular adverse events occurring in patients receiving anticancer medications, especially for newer drugs that are able to provide better cancer survival and that showed to be better tolerated by the ocular surface than traditional anticancer drugs. Although pathological mechanisms have been identified or in some instances hypothesized, larger case-controlled studies are required to further elucidate the impact of ocular surface side effects on the clinical course of the oncological patient and hopefully to identify potential predictors for these occurrences.”

Furthermore, we revised the English throughout the manuscript.